# Emerging Therapeutic Potential of Fluoxetine on Cognitive Decline in Alzheimer’s Disease: Systematic Review

**DOI:** 10.3390/ijms25126542

**Published:** 2024-06-13

**Authors:** Anastasia Bougea, Efthalia Angelopoulou, Efthimios Vasilopoulos, Philippos Gourzis, Sokratis Papageorgiou

**Affiliations:** 11st Department of Neurology, “Aiginition” Hospital, School of Medicine, National and Kapodistrian University of Athens, 11528 Athens, Greece; angelthal@med.uoa.gr (E.A.); sokpapa@med.uoa.gr (S.P.); 2First Department of Psychiatry, “Aiginition” Hospital, School of Medicine, National and Kapodistrian University of Athens, 11528 Athens, Greece; efvasilop@med.uoa.gr (E.V.); pgourzis@upatras.gr (P.G.); 3Department of Psychiatry, University of Patras, 26504 Patras, Greece

**Keywords:** fluoxetine, Alzheimer’s disease, cognitive decline, BDNF, neurogenesis

## Abstract

Fluoxetine, a commonly prescribed medication for depression, has been studied in Alzheimer’s disease (AD) patients for its effectiveness on cognitive symptoms. The aim of this systematic review is to investigate the therapeutic potential of fluoxetine in cognitive decline in AD, focusing on its anti-degenerative mechanisms of action and clinical implications. According to PRISMA, we searched MEDLINE, up to 1 April 2024, for animal and human studies examining the efficacy of fluoxetine with regard to the recovery of cognitive function in AD. Methodological quality was evaluated using the ARRIVE tool for animal AD studies and the Cochrane tool for clinical trials. In total, 22 studies were analyzed (19 animal AD studies and 3 clinical studies). Fluoxetine promoted neurogenesis and enhanced synaptic plasticity in preclinical models of AD, through a decrease in Aβ pathology and increase in BDNF, by activating diverse pathways (such as the DAF-16-mediated, TGF-beta1, ILK-AKT-GSK3beta, and CREB/p-CREB/BDNF). In addition, fluoxetine has anti-inflammatory properties/antioxidant effects via targeting antioxidant Nrf2/HO-1 and hindering TLR4/NLRP3 inflammasome. Only three clinical studies showed that fluoxetine ameliorated the cognitive performance of people with AD; however, several methodological issues limited the generalizability of these results. Overall, the high-quality preclinical evidence suggests that fluoxetine may have neuroprotective, antioxidant, and anti-inflammatory effects in AD animal models. While more high-quality clinical research is needed to fully understand the mechanisms underlying these effects, fluoxetine is a promising potential treatment for AD patients. If future clinical trials confirm its anti-degenerative and neuroprotective effects, fluoxetine could offer a new therapeutic approach for slowing down the progression of AD.

## 1. Introduction

By affecting 47 million people worldwide, Alzheimer’s disease (AD) is the commonest progressive neurodegenerative disorder, characterized by cognitive decline and behavioral and mood disorders, with an enormous socioeconomic burden [1,2]. The hallmarks of AD are the accumulation within neurons of neurofibrillary tangles, resulting from the hyperphosphorylation of the cytoskeletal tau protein, and the formation of senile plaques generated from the extracellular accumulation of the amyloid beta peptide (Aβ); this is the product of an altered metabolism of the human amyloid precursor protein (APP), which is cleaved and accumulates as neurotoxic amyloid plaques within the extracellular space of AD brains. Despite encouraging results in targeting the accumulation of amyloid beta pathology by novel medications utilizing anti-amyloid beta monoclonal antibodies, there is currently no definite cure for AD [3]. There are symptomatic treatments that alleviate cognitive symptoms only for a short time, without slowing down the progression of the disease [4]. There is an unmet need to develop an alternative therapeutic strategy for dementia.

Acetylcholine (Ach) deficiency is considered to be one of key factors underlying cognitive impairment in AD [5]. Several studies suggest that sigma receptor ligands can elevate Ach extracellular levels in prefrontal cortex rat brain [6,7]. Given the high sigma-1 receptor affinity of fluoxetine, this selective serotonin reuptake inhibitor (SSRI), which is commonly used to treat depression, attracts attention for managing cognitive deficits in AD [8,9,10]. Indeed, there is a growing evidence showing the neuroprotective effect of fluoxetine against amyloid beta Aβ plaques and hyperphosphorylated tau tangles, as the hallmark pathological features of AD and tau pathology [11]. In addition, common pathophysiological events have been identified in depression and AD, including neuroinflammation with aberrant Tumor Necrosis Factor-α (TNF-α) signaling, and an impairment of Brain-Derived Neurotrophic Factor (BDNF) and Transforming-Growth-Factor-β1 (TGF-β1) signaling. Fluoxetine reduced neuroinflammation and restored neuroplasticity by regulating the aforementioned signaling pathways in animal AD models. Since clinical trials are lacking, the role of fluoxetine in cognitive function in people with AD has not yet reached a consensus in clinical practice.

Previously, four meta-analyses investigated the efficacy of SSRIs, including fluoxetine, on cognitive performance in dementia non-AD (vascular dementia, frontotemporal dementia) [12], AD [13,14], and an unspecific dementia diagnosis [15], with conflicting outcomes. This systematic review aims to evaluate the existing evidence on the therapeutic potential of fluoxetine in cognitive impairment in AD, focusing on its mechanisms of action, preclinical and clinical evidence, and implications for future research and clinical practice.

## 2. Methods

### 2.1. Search Strategy

For the purposes of this systematic review, we followed the PRISMA 2020 principles [16]. The PubMed and MEDLINE databases were searched for peer-reviewed research articles investigating and/or discussing the efficacy of fluoxetine treatment for cognitive symptoms of AD, published in the English language with no time restrictions. The search was conducted between September 2022 and April 2024. Both preclinical and clinical studies were included. In order to include most of the key recent evidence, we screened the references of the selected articles for additional relevant articles. We used the terms “Alzheimer’s disease”, “cognitive decline”, “memory loss”, “treatment”, “therapy”, “pharmacological”, and “Fluoxetine” in different combinations. For the narrative synthesis, our initial categorization was based on the efficacy of fluoxetine having been investigated in cognitive function of AD animal and human studies, followed by a discussion of limitations and potential future directions. Figure 1 shows the flow diagram that depicts the process of study selection.

### 2.2. Eligibility Criteria

Two authors (A.B. and E.A.) independently assessed all titles and abstracts, obtained full texts for potentially relevant studies, and applied the following inclusion criteria (Table 1).

### 2.3. Study Selection and Data Extraction

Two authors (A.B. and E.A.) independently evaluated the titles and abstracts of studies that potentially met the inclusion criteria and resolved any uncertainty through discussion. Data were extracted independently by two reviewers (A.B. and E.A.) and entered onto a standardized, pre-piloted data extraction form for assessment of study quality and evidence synthesis. The extracted characteristics were study/year, study population, FLX dosage, cognitive measurements (biomarkers, tests), and the effect of FLX on AD animal models. If data were missing in an original study for certain outcomes, the method of imputation was used.

## 3. Results

### 3.1. Articles Included in Search

From 854 papers, after eliminating 642 duplicates and rejecting particular studies according to the exclusion criteria, 129 papers were reviewed based on the inclusion criteria. Finally, 22 studies (19 animal and 3 clinical trials) were included (Figure 1). Due to the heterogeneity of the studies, meta-analysis was not performed.

From the nineteen animal AD studies, nine studies used APP/PS1 mice as the AD model, two studies used the C57BL/6 non-Tg model of AD, and one study used the AD Tg2576 model. Eight studies used male mice. Eleven studies evaluated the spatial and learning memory of AD animals using the following tests: the Y Maze and Water Maze Test. Nineteen studies were analyzed based on the therapeutic potential of fluoxetine in cognitive function of AD models, exploring anti-degenerative, anti-inflammatory, and neuroprotective mechanisms, as follows:

### 3.2. Mechanisms of Anti-Degenerative Action of Fluoxetine—Evidence from Anmals Studies

#### 3.2.1. Modulation of Amyloid Beta and Tau Pathology

Fluoxetine modulated Aβ and tau pathology in preclinical models of AD by reducing the production of Aβ products, in part via the DAF-16-mediated cell signaling pathway [17,18,19,20,21,22,23], and inhibiting tau hyperphosphorylation. DAF-16 regulates the process that convert toxic oligomers to less-toxic aggregates [23]. In this way, fluoxetine exerts a protective effect, which is further supported by the translocation of DAF-16 from the cytosol to the nucleus [23]. Nevertheless, how DAF-16 mediates the protection by fluoxetine against Aβ toxicity remains obscure. Fluoxetine also reduced Aβ toxicity by activating Wnt/β-catenin signaling, which in turn downregulated the expression of BACE1 in the hippocampus of 3×Tg AD mice [24]. This suggests that there might be an association between the neuroprotective effect of fluoxetine and the regulation of the Wnt signaling pathway in the AD brain. Fluoxetine had neuroprotective properties against the synaptotoxic effects of Aβ oligomers by increasing TGF-β1 [25]. Interestingly, this study concluded that the deficit of hippocampal TGF-β1 is a long-lasting molecular marker associated with a depressive-like phenotype and memory deficits in a non-Tg model of AD. Fluoxetine increased the intracellular levels of the precursor of TGF-β1 (Pro-TGFβ1) without affecting TGFβ1 mRNA expression. In addition, we demonstrate that fluoxetine affected the conversion of Pro-TGF-β1 into active TGF-β1, likely through the activation of MMP-2. The involvement of MMP-2 in fluoxetine activity is in line with the suggestion that extracellular matrix-modifying enzymes contribute to antidepressant-mediated structural plasticity in the hippocampus [26]. TGF-β1 is an anti-inflammatory cytokine that exerts neuroprotective effects in different models of amyloid-induced neurodegeneration [27]. Four studies observed that fluoxetine-treated mice and Aβ rats performed better in spatial learning, working, and reference memory as measured by MWM, Ymaze [20,22,24,28,29]. Fluoxetine also improved more specific cortical cognitive functions of cholinergic nucleus basalis Meynert (NBM)-lesioned rats (experimental model of AD) [30]. The NBM has major efferent projections to medial temporal structures, the amygdala, frontoparietal cortex, and temporo-parietal association areas [31]; thus, stimulation of these pathways by fluoxetine might enhance cholinergic activity in these cortical regions and thereby improve mnemonic, attentional, and perceptive abilities, respectively. The acute administration of fluoxetine has been reported to reverse the depressive-like effect induced by Aβ1-40 administration [32]. Robust Aβ1-40 immunoreactivity was verified using an anti-oligomer monoclonal antibody (NU4)6 in hippocampi from Aβ1-40-injected mice, but not in hippocampi from control vehicle-injected animals. Together, these results indicate that Aβ1-40 has an acute impact on memory, learning, and mood in mice, and that fluoxetine treatment prevented both the cognitive impairment and depressive-like behavior induced by Aβ1-40. By targeting these key protein aggregates, fluoxetine may help to mitigate neuronal dysfunction and slow the progression of neurodegeneration in AD.

#### 3.2.2. Neuroprotection against Neuroinflammation and Oxidative Stress

##### Modulation of Microglial Activation

Microglia, the resident immune cells of the central nervous system, play a crucial role in regulating neuroinflammation in AD [33]. Dysregulated microglial activation can lead to the release of pro-inflammatory cytokines and the production of reactive oxygen species, contributing to neuronal damage. Fluoxetine has been shown to modulate microglial activation by enhancing the release of TGF-β1 from microglial cells [34]. Pro-inflammatory bacterial lipopolysaccharide (LPS) stimulation, a widely used model of inflammation, drives microglial activation through the Toll-Like Receptor (TLR)-4 and nuclear factor kB (NF-κB) pathways, triggering transcriptional activation of various microglial phagocytic markers, growth factors, and cytokines [35]. By inhibiting LPS-induced microglial activation and the subsequent release of pro-inflammatory factors such as TNFα, IL-1β, fluoxetine exerted neuroprotection against microglia-mediated neurotoxicity. Thus, fluoxetine not only can relieve depression, but it might also hold the potential to retard the inflammation-mediated chronic neurodegenerative process of AD.

##### Regulation of the NF-κB Signaling Pathway

The nuclear factor-kappa B (NF-κB) signaling pathway is a key regulator of inflammation and immune responses in the brain. Dysregulation of NF-κB activity has been implicated in the pathogenesis of AD, leading to excess production of pro-inflammatory mediators [36]. NF-κB activates BACE1, which promotes Aβ fibril formation [37]. Consequently, Aβ fibrils directly activate NF-κB, leading to the expression of APOE4 and mGluR5. In microglia and astrocytes, Aβ42 activates NF-κB, which induces the expression of proinflammatory factors that cause myelin injury. Additionally, NF-κB activates microRNA production (such as miRNA-125b, miRNA-9, miRNA-155, miRNA-34a, and miRNA-146a), which suppresses the expression of various neuroprotective factors (neuroprotectin D1, vitamin D3 receptor) [38]. Similarly, the formation of hyperphosphorylated tau in AD brain is enhanced by the NF-κB-dependent activation of SET, which inhibits tau’s dephosphorylation [39]. Conversely, glycated tau triggers ROS production, leading to NF-κB activation. Collectively, the different pathways that contribute to neurodegeneration in AD are highly interconnected via continuous NF-κB activity in both neurons and glial cells. Fluoxetine may inhibit NF-κB signaling by preventing the nuclear translocation of NF-κB/TLR4/NLRP3 and suppressing the expression of pro-inflammatory genes in AD models [40]. In addition, it was shown that the decrease in the expression of TLR4, NF-κB, NLRP3, caspase-1, TNF-α and IL-1β caused by fluoxetine treatment (10 mg/kg/day, p.o.) attenuated the progression of Alzheimer-like phenotypes in socially isolated depressed-like rats [40]. By targeting the NF-κB pathway, fluoxetine may exert anti-inflammatory effects and mitigate neuroinflammation in AD.

##### Neuroprotective Effects on Astrocytes

Astrocytes, the most abundant glial cells in the brain, play a critical role in supporting neuronal function and synaptic transmission [41]. Dysfunctional astrocytes produced inflammatory cytokines such as TNF-α, IL-1β and inducible nitric oxide synthase (iNOS), which in turn produce neurotoxic free radicals ROS (nitric oxide) [42]. Excessive ROS production can lead to oxidative damage to cellular components, including proteins, lipids, and DNA, contributing to neuronal degeneration. Fluoxetine has been shown to reduce inflammation and oxidative stress in the brain, via targeting antioxidant Nrf2/HO-1 and hindering the TLR4/NLRP3 inflammasome signaling pathways [40]. Fluoxetine has been proved to possess valuable antioxidant properties by reducing the expression of pro-oxidant iNOS and Nox2 enzymes in an AD non-transgenic model [34]. By protecting against oxidative stress, fluoxetine may help to preserve neuronal integrity and prevent both cognitive deficits and a depressive-like phenotype [34]. Fluoxetine protected the hippocampal neurons by inhibiting the activation of astrocytes, reducing Aβ products [43]. These promising results are the basis for the clinical application of fluoxetine as a novel astrocyte-based therapy of AD.

#### 3.2.3. Promotion of Neurogenesis and Synaptic Plasticity

Neurogenesis is the process of generating new neurons from neural stem cells. Synaptic plasticity is the ability of synapses to adapt and strengthen in response to stimuli. These neural processes play crucial roles in maintaining brain function and cognitive abilities [44]. Impaired hippocampal neurogenesis and synaptic dysfunction are common features of AD, contributing to cognitive impairment [45]. Fluoxetine increased the dendritic density in the CA1/2 and CA3 regions of the hippocampus, enhancing the learning memory in a transgenic AD model [18,21,28,46]. Moreover, fluoxetine may promote hippocampal neuroplasticity and memory by enhancing ILK-AKT-GSK3β pathway activity in APP/PS1 mice [47]. Activated Caspase-3 prevented tau phosphorylation in a transgenic AD mouse model, via the Akt- p-GSK3β levels, suggesting a potential therapeutic effect for AD [48]. Moreover, elevating ILK protein expression in the hippocampus could restore impaired AKT-GSK3β pathway activity in APP/PS1 mice.

Brain-Derived Neurotrophic Factor (BDNF) is a neurotrophic factor that plays a crucial role in the growth and maintenance of neurons. Fluoxetine increased p-CREB and BDNF levels in the hippocampus of 3×TgAD mice via the activation of the CREB/p-CREB/BDNF signaling pathway [17,20,40]. p-CREB and BDNF promoted neuroplasticity, including the enhancement of hippocampal and cortex volume, increased numbers of neurons, remodeled dendritic plasticity, and increased synaptic protein levels (such as GluN2A, GluN2B, GluA1, GluA2, PSD93, PSD95, SYN1, and SYT). In addition, this increased expression of BDNF may also contribute to the potentiation of LTP in 3×TgAD mice. Fluoxetine administration during adolescence produced longer-lasting and higher beneficial effects on the expression of BDNF, and its regulatory molecules’ Aβ-oligomers reduce BDNF signaling by impairing the axonal transport of BDNF in the neurons of AD transgenic mice (Tg2576). The deficit of BDNF can induce an abnormal accumulation of Aβ and synaptic dysfunction, finally leading to cognitive decline. In this context, the presence of the functional single-nucleotide polymorphism (Val66Met), which impairs BDNF signaling, significantly increases the risk of developing depression in AD patients, thus suggesting that a deficit of BDNF release might be a common pathophysiological event both in depression and AD. However, the underlying mechanisms need further analysis. On the other hand, another study failed to demonstrate a positive effect of fluoxetine on neurogenesis and enhanced BDNF protein in the hippocampus of 3×Tg mice, suggesting a more complex role of environmental factors for neurogenesis [49].

Overall, preclinical studies suggested that fluoxetine improved cognitive function in animal models of AD through the modulation of amyloid beta/tau pathology, neuroplasticity, and neuroinflammation. The evidence from AD animal studies is summarized in Table 2.

### 3.3. Evidence from Clinical Trials

While preclinical studies have provided compelling evidence of the anti-degenerative effects of fluoxetine in AD, clinical trials investigating its therapeutic potential are largely limited (Table 3). Only three clinical trials have investigated the effects of fluoxetine in AD patients. For example, a pilot study by Mowla et al. [50] suggested positive effects of fluoxetine on the cognitive measures, daily living, and global functioning of a small group of AD patients, especially in combination with cholinesterase inhibitors. Similarly, a double-blind randomized controlled trial by Taragano et al. [51] found that fluoxetine improved the depressive mood and cognitive scores of AD patients. However, the quality of this study was impacted by the relatively short treatment duration, small sample size, and concomitant major depression (selection bias), emphasizing the importance for larger and longer well-designed clinical trials on the efficacy of fluoxetine for AD. By contrast, Petracca et al. [52] revealed no benefit on the cognitive performance of 41 patients with AD.

### 3.4. Quality Assessment

Two reviewers (A.B. and E.A.) evaluated the quality of animal studies using the ARRIVE tool for assessing risk [53]. In cases of disagreement, these were resolved by discussion. The ARRIVE tool includes the following domains: random sequence generation, allocation concealment, blinding of participants and personnel, blinding of outcome assessment, incomplete outcome data, selective reporting, and other bias. Each domain was evaluated with “high risk of bias”, “low risk of bias” (Table 4). Overall, animal studies showed a high quality.

Regarding the three clinical trials, we used the Cochrane tool for assessing risk of bias in randomized clinical trials [54]. The Cochrane tool is based on seven bias domains: (i) sequence generation (selection bias), (ii) allocation concealment (allocation bias), (iii) blinding of participants and personnel (performance bias), (iv) blinding of outcome assessors (detection bias), (v) incomplete outcome data (attrition bias), (vi) selective reporting (reporting bias), and (vii) an auxiliary domain: “other bias”. For each bias domain, we assigned a score of a “high”, “low”, or “unclear” risk of bias (Table 5). All three trials were of low to moderate quality.

## 4. Discussion

There is limited evidence of the effectiveness of fluoxetine on the cognitive function of people with AD (only three clinical trials). Preclinical studies of a high quality suggested that fluoxetine repaired the cognitive function of animal models of AD, promoting neuroplasticity against neuroinflammation and oxidative stress (Table 1 and Table 2). However, results from animal studies do not always translate to humans, and further research is warranted to confirm the efficacy and safety of fluoxetine in clinical AD settings.

Although the clinical studies were double-blind RCTs, there was evidence of an unclear risk of bias in some domains, such as incomplete cognitive outcome data [50,51,52]. According to the ARRIVE system, we classified the overall quality of evidence as ‘low to moderate’ for both treatment response and cognitive symptoms, due to methodological flaws.

We used a comprehensive and sensitive strategy to identify animal and human studies reported in the literature. Following selection of the studies, data extraction and assessments of risk of bias were independently conducted by two authors. However, there are limitations to our review. Identified RCTs were heterogeneous with regard to the participants selected. Moreover, in one of the trials, participants had concomitant major depression [51]. The duration of exposure to fluoxetine also differed, and these studies had fewer than 50 participants.

### 4.1. Correlation between Neuroprotective/Antioxidant/Anti-Inflammatory Properties and Antidepressant Activity: Dual Action of Fluoxetine

There is a crosslink between the neuroprotective/antioxidant/anti-inflammatory properties and antidepressant activity of fluoxetine. Depression and AD may share common mechanisms, including neuroinflammation with aberrant Tumor Necrosis Factor-α (TNF-α) signaling, and an impairment of Brain-Derived Neurotrophic Factor (BDNF) and Transforming-Growth-Factor-β1 (TGF-β1) signaling. The antidepressant fluoxetine showed neuroprotection in AD rats by improving their depressive behavior, as well as their learning memory and cognition. The fluoxetine treatment of depressive and AD animals showed that it (i) enhanced animals’ brain monoamines of 5HT, DA, and NE, with a decrease in AChE activity; (ii) evoked neurogenesis, indicated by BDNF enhancement; (iii) reduced oxidative stress, manifested by the upregulation of Nrf-2/HO-1 expression, with increased SOD and TAC content, along with decreased MDA content; (iv) decreased neuroinflammation, reported by the suppression of the protein expression of the NF-κB/TLR4/NLRP3 pathway as it downstreamed effectors, namely, TNF-α, IL-1β content; and (v) decreased procaspase 1 protein expression. Consequently, it decreased brain β-amyloid and tau protein, as well as exhibiting mild to moderate histopathological features of brain histopathology, and decreasing neuronal degeneration and preventing neuronal pyknosis. Zhao et al. [55] reported the neuroprotective potential of fluoxetine against neural inflammation and apoptosis in chronic stressed animals. Moreover, Du et al. [56] showed the neuroprotective role of fluoxetine in inhibiting NLRP3 inflammasome activation in depressed animals. In addition, Mendez-David et al. [57] investigated the neuroprotective activity of chronic fluoxetine treatment in a mouse model of anxiety/depression via modulating Nrf2 and BDNF signaling. These findings may open new horizons for fluoxetine as an anti-depressive drug with the potential to eliminate the risk of AD. While there is evidence that the antidepressant effect of fluoxetine requires its pro-neurogenic action, exerted by promoting the proliferation, differentiation, and survival of the progenitor cells of the hippocampus, on the other hand, fluoxetine in combination with exercise also exerts neurogenesis-independent antidepressant effects by influencing the plasticity of the new neurons generated [58].

In isolated mitochondria, fluoxetine indirectly affects electron transport and (F1Fo) ATPase activity and thus inhibits oxidative phosphorylation, as a result of interfering with the physical state of the lipid bilayer of the inner mitochondrial membrane. Moreover, fluoxetine promoted mitophagy by the increased colocalization of autophagosomes and mitochondria, eliminating damaged mitochondria in corticosterone-treated astrocytes [59]. A further in vitro study showed that p53 presence is required for fluoxetine-activated autophagy flux, and fluoxetine promotes astrocytic autophagy in a p53-dependent mechanism. These results may be the first step towards a novel approach to treat depression depending on astrocytes, and it provides a promising molecular target for the development of antidepressant drugs, besides regulating neurotransmitters.

### 4.2. Implications for Practice and Research

Fluoxetine exerts its beneficial cognitive effects via neuronal signaling mechanisms potentially involved in restoring neuroplasticity neuroinflammation.

Only three clinical studies have so far evaluated the cognitive effects of fluoxetine in patients with AD [50,52]. Further well-designed multicenter RCTs should adhere to high standards of methodology and reporting and include diagnostically homogenous populations and large samples. Given the lack of rigorous evidence, it is difficult to form an evidence-based policy whether fluoxetine is effective in treating cognitive deficits in people with AD. Future research should focus on establishing the optimal dosing, duration of treatment and patient characteristics that may influence the response to fluoxetine. Longitudinal studies are needed to demonstrate the disease-modifying activity of fluoxetine and delay cognitive decline in preclinical AD by the rescuing of TGF-β1 signaling.

In addition to clinical trials, translational research bridges the gap between preclinical and clinical findings. Future imaging studies using techniques such as functional magnetic resonance imaging (fMRI) and amyloid β and amyloid pathology observed on positron emission tomography (PET) scans would help elucidate the neural networks and neurotransmitter systems affected by fluoxetine in AD patients. Nevertheless, the practical application of these biomarkers faces problems stemming from concerns about their invasiveness, considerable expense, and the limited accessibility of PET imaging.

Clinical practice guidelines for the use of fluoxetine in cognitive impairment in AD should be informed by the latest evidence from research studies. Healthcare providers should consider the potential benefits and risks of fluoxetine on an individual basis, taking into account factors such as medical history, comorbidities, and medication interactions. Close monitoring of cognitive function, mood, and overall well-being is essential during fluoxetine treatment to assess treatment response and tolerability. For this purpose, it is crucial to perform longitudinally a neuropsychological and neuroimaging evaluation, along with the monitoring of markers indicating neuronal loss and assessments of brain metabolism as Aβ amyloid and tau protein.

### 4.3. Challenges and Limitations

Despite promising outcomes from animal studies and early clinical trials, there are several challenges and limitations that need to be considered when evaluating the efficacy of fluoxetine for AD. One major challenge is the heterogeneity of AD as a clinical syndrome, with variable cognitive profiles and disease progression, which may complicate the assessment of treatment effects. Factors such as study design, patient characteristics, and treatment duration may all impact the outcomes of studies on fluoxetine in this population. Additionally, individual differences in response to medication and the presence of other medical conditions or medications influence the effectiveness of fluoxetine in managing depression in individuals with AD. This variability can make it difficult to determine the specific effects of fluoxetine on cognitive function and other symptoms of the disease. Regarding clinical trials, small sample sizes, short treatment durations, and variations in study design can influence the safety and efficacy of fluoxetine on cognitive decline. Future research should aim to address these limitations and clarify the potential benefits of fluoxetine in AD.

A significant limitation is the translation of preclinical findings to human clinical trials. While several preclinical studies have shown positive results [17,20,25,34], it is essential to determine whether fluoxetine can effectively reduce beta-amyloid levels in human patients with AD and whether this reduction translates into clinical benefits such as cognitive improvement and disease modification. Furthermore, the optimal dosing and treatment duration of fluoxetine for targeting beta-amyloid in AD need to be further elucidated. Most animal AD studies have employed relatively short-term fluoxetine treatment regimens, and it is unclear whether long-term or chronic fluoxetine administration may have differing effects on beta-amyloid pathology. Notably, enhancing the neurogenesis of the dentate gyrus may be a mechanism by which the severity of cognitive deficits in AD patients can be improved [21]. Future research should explore how cell proliferation or survival with the dentate gyrus is influenced by multiple factors, such as drugs, stress, exercise, and the multiplicity of environmental inputs.

Another weakness is the potential for drug interactions and neuropsychiatric side effects [60]. The most frequent adverse effects, which occurred in 10–25% of patients, were nausea (25%), nervousness, insomnia, headache, tremor, anxiety, drowsiness, dry mouth, sweating, and diarrhea (10%). Most of these adverse effects occurred early in treatment and seldom led to drug withdrawal. Fluoxetine worsened parkinsonian disability and akathisia because of the enhanced serotonergic inhibition of dopamine neurons [61]. Fluoxetine has been associated with seizures, both in therapeutic doses and in overdose [62,63]. There was an increased rate of suicide attempts, acute mania, and manic-like behavior after treatment with fluoxetine [64]. Interestingly, cognitive function can be impaired by fluoxetine; a negative effect on learning and memory has been described. These side effects can be particularly problematic in elderly patients with multiple comorbidities, who may be more vulnerable to the adverse effects of medication. Additionally, fluoxetine can interact with other medications commonly used in AD patients, such as cholinesterase inhibitors and memantine, which may complicate treatment regimens and increase the risk of side effects [65].

Furthermore, the optimal dose and duration of fluoxetine treatment for cognitive decline in AD are not well established. One study used higher doses of fluoxetine than those typically used for depression, suggesting that higher doses may be necessary to achieve neuroprotective effects in AD patients [52]. However, higher doses of fluoxetine can also increase the risk of side effects and drug interactions, highlighting the need for careful dosing and monitoring in clinical practice.

## 5. Conclusions

This is an up-to-date, methodologically rigorous systematic review on the efficacy of fluoxetine for cognitive impairment in patients with AD, including animal and human studies. We found no clear evidence to support the efficacy of fluoxetine for treating cognitive deficits in AD. Fluoxetine shows promise as a potential treatment for AD, based on its neuroprotective and anti-inflammatory effects observed in animal studies. However, there are several challenges and limitations that need to be addressed before fluoxetine can be considered as a standard treatment for AD. These findings do not mean that people with severe depression and AD should not be treated with antidepressants. Given this review is limited by a paucity of trials, small studies overall, and a variation in patients recruited, further RCTs are needed to confirm the effects in AD. Larger and longer clinical trials are needed to further evaluate the efficacy of fluoxetine in different subtypes of AD patients and to establish the optimal dose and duration of treatment. Additionally, more research is needed to investigate the potential mechanisms of action of fluoxetine in AD and to identify biomarkers that can predict treatment response. Overall, the evidence for the efficacy of fluoxetine for AD is promising but preliminary, and further research is warranted to fully assess its potential benefits in patients with this devastating condition.

## Figures and Tables

**Figure 1 ijms-25-06542-f001:**
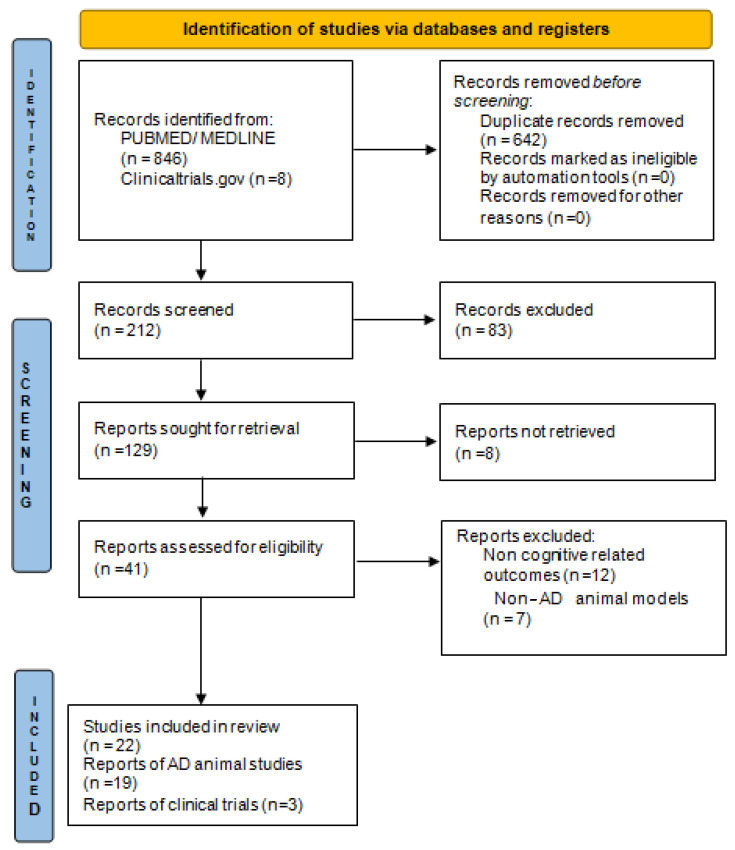
PRISMA 2020 flowchart of study selection.

**Table 1 ijms-25-06542-t001:** Eligibility criteria of included studies.

	Inclusion Criteria	Exclusion Criteria
Study type	Clinical trials Animal studies Cellculture studies	systematic reviews, letters to the editor, editorials, conference abstracts, and theses duplicate reports of the same population. Ongoing studies, studies not available in English, and unpublished studies
Study sample	AD diagnosis by validated criteria	mild cognitive impairment vascular dementia, frontotemporal dementia, and/or delirium without a distinct dementia
Study intervention	Placebo-controlled studies of FLU	studies that referred only to other antidepressants or used comparisons with other alternative active treatment
Study outcomes	Cognitive performance assessed by a validated cognitive test	non-AD-related cognitive or behavioral endpoints alone

**Table 2 ijms-25-06542-t002:** Evidence from AD animal studies for effectiveness of fluoxetine in cognitive performance.

[Ref]	Study Population	Route, Dose of FLU	Cognitive Measurements (Biomarkers, Tests)	CognitiveEffect of FLU on AD Animal Models
[30]	adult male Wistar rats: intact control sham-operated NBM lesioned	3, 5, and 10 mg/kg	active avoidance	improved learning and memory processes in NBM-lesioned rats
[21]	Tg2576 (AD model) and control mice	5 mg/kg or saline vehicle	i.p. 150 mg/kg BrdU Behavioral testing of contextual memory, cued memory, and cued fear	(1) increased cell proliferation in the dentate gyrus (2) improved contextual memory
[23]	*C. elegans* strain CL2006 model of Aβ toxicity wild/double transgenic APP/PS1 mice (positive controls)	1 mM	thermal stress resistance test	delayed Abeta-induced paralysis in the *C. elegans* model
[32]	AβO-injected mice and vehicle-injected control mice	30 mg kg^−1^ or salin	amyloid-β peptide Porsolt FST tail suspension test (TST) object recognition task	anti-inflammatory effect in AβO-injected group
[49]	3×Tg mice	18 mg/kg/day per.os	BrdU BDNF protein WMS	no effect on long-term survival of newborn neurons
[22]	APP/PS1 mice	5 mg/kg/day	brain tissue, CSF, sera blood-soluble β-amyloid	(1) prevented the SYP, MAP2 protein loss (2) improved spatial memory, learning (3) decreased brain, CSF, sera Aβ levels
[43]	APP/PS1 mice	5 mg/kg/day	Aβ40 and Aβ42 WMS Y Maze Test	(1) inhibited activation of astrocytes (2) lowered Aβ (3) attenuated neurotoxicity (4) improved behavioral performance
[17]	6-month-old 3×TgAD mice	i.p. 1 mL per 100 g or vehicle	WMS fear-conditioning tests	reduced beta-amyloid levels increased the hippocampal CA1 and dentate gyrus volumes
[19]	APP/PS1 transgenic AD model mice	i.p. 10 mg/kg/day	WMS	prevents the loss of DG neurons but not in the CA1 and CA3 of the hippocampus
[20]	adult 3×TgAD mice and male non-transgenic wild-type control mice	i.p. 10 mg/kg body weight	WMS	(1) increased the sizes of the hippocampal CA1, DG, and extensive cortex regions (2) improved cognitive deficits
[47]	C57BL/6J and APP/PS1 mice	NA	ILK WMS contextual fear conditioning novel ORT	improved the hippocampal neurogenesis and memory
[24]	36 APP/tau/PS1 mice: 12Tg + FLX 12Tg 12 WT	intragastric 20 mg/kg	brain tissue Aβ1-42 and Aβ40 levels WMS spatial learning test probe trial	(1) reversed spatial learning deficits (2) improved short-term and long-term memory
[18]	40 8-month-old male APP/PS1 double-transgenic AD mice	10 mg/kg/day	brain tissue Aβ40 and Aβ42 WMS	(1) decreased soluble Aβ40 and Aβ42 (2) delayed dendritic spine synapse loss of hippocampus (3) improved learning ability during early AD
[25]	male C57BL/6 mice (non-Tg model of AD): 1st cohort: FLU, VTX 2nd cohort: PBS i.c.v. + vehicle i.p., Aβ i.c.v. + vehicle i.p., Aβ i.c.v. + FLU., Aβ i.c.v. + VTX 5 mg/kg i.p., and Aβ i.c.v. + VTX 10 mg/kg i.p. 3rd cohort: 2nd cohort + FLU 10 mg/kg, VTX 5 mg/kg, and VTX 10 mg/kg	1st cohort: 10 mg/kg i.p. 2nd cohort: 10 mg/kg i.p. 3rd cohort: 20 mg/kg	FST ORT PAT	reversed depressive-like phenotype and memory deficits induced by Aβ1-42 oligomers
[46]	middle-aged APP/PS1 mice	NA	PSD-95 SYN-1	(1) increased no of dentritic hypocambal spines (2) enhanced learning ability
[28]	29 8-month-old male APP/PS1 Mice	i.p. 10 mg/kg/day	WMS Y Maze Test Open Field Test	(1) improves memory and learning tasks (2) delays senescence of oligodendrocyte lineage cells in hipocampus
[29]	adult male Wistar rats: 6LD, 6LL, 6Aβ, 6Aβ(s) 6LL + Aβ(s)	5 mg/kg/day	WMS	improved memory performance, oxidative damage, and amyloid aggregation
[40]	12NC 12SI 4SI (FLU) 4AD, 4AD (FLU) 4SI + AD	10 mg/kg/day, p.o.	FST WMS	increased serum levels of HDL and reduced serum cholesterol levels decreased brain β-amyloid and tau protein
[34]	male C57BL/6 mice: (1) PBS + Veh; (2) Aβ + Veh; (3) Aβ + FLU; (4) Aβ + VTX 5 mg/kg; (5) Aβ + VTX 10 mg/kg	i.p. 10 mg/kg	FST ORT PAT	reversed Aβ-induced neurodegeneration in mixed neuronal cultures treated with Aβ oligomers

Aβ: amyloid beta protein, APP/PS1: amyloid precursor protein/presenilin 1, BrdU: 5-Bromodeoxyuridine, DG: dentate gyrusfluoxetine, FLU: fluoxetine, FST: forced swim test, i.p.: intraperitoneal, LD: light–dark, LL: constant light, NA: non-available, NBM: nucleus basalis Meynert, NC: normal control, MAP2: microtubule associated protein 2, ORT: object recognition test, PAT: Passive Avoidance Test, PSD-95: postsynaptic density 95, SI: social isolation, SYN-1: synapsin-1, SYP: synaptophysin, TST: tail suspension test, WMS: Morris Water Maze Test, Veh: vehicle, VTX: vortioxetine.

**Table 3 ijms-25-06542-t003:** Evidence from clinical trials for effectiveness of fluoxetine.

[Ref]	Study Type	Study Population	FLU Alone or in Combination with Other Agents	Cognitive Measurements (Biomarkers, Tests)	Effect of FLU on AD Patients
[52]	double-blind, parallel design	41 AD pts	up to 40 mg/day or placebo	MMSE HAM-D CGI HAM FIM	no effect on MMSE scores
[50]	12 weeks randomized, placebo-controlled, double-blind	42 pts mild-to-moderate AD (55–85 age)	41 FLU + R: 20 mg/day + 6–12 mg/day 41 R group: 6–12 mg/day 40Placebo	MMSE WMS-III ADL HAM CGI	improvement in measures of cognition and memory
[51]	randomized, fixed-dose, double-blind	25 pts with AD and major depression (mean age 71.7)	AMIT group: 25 mg/day FLU group: 10 mg/day over the 6-week period.	MMSE HAM	improvement in MMSE scores

ADL: activities of daily living, AMIT: Amitriptyline, FIM: Functional Independence Measure, CGI: Clinical Global Impression, HAM: Hamilton Rating Scale for Anxiety, HAM-D: Hamilton Depression Scale-17, MMSE: Mini-Mental Status Examination, R: rivastigmin, WMS-III: Wechsler Memory Scale–3rd Edition.

**Table 4 ijms-25-06542-t004:** Quality of animal studies according to ARRIVE assessment tool.

[Ref]	Ethical Statement	Study Design	Experimental Procedures	Experimental Animals	Housing and Husbandry	Sample Size	Allocating Animals to Experimental Groups	Experimental Outcomes	Statistical Analysis
[30]	+	+	+	+	+	+	+	+	+
[21]	+	+	+	+	+	+	+	+	+
[23]	+	+	+	+	+	+	+	+	+
[32]	+	+	+	+	+	+	+	+	+
[49]	+	+	+	+	+	+	+	+	+
[22]	+	+	+	+	+	+	−	+	+
[43]	+	+	+	+	+	+	−	+	+
[17]	+	+	+	+	+	+	−	+	+
[19]	+	+	+	+	+	+	−	+	+
[20]	+	+	+	+	+	+	+	+	+
[47]	+	+	+	+	+	+	+	+	+
[24]	+	+	+	+	+	+	+	+	+
[18]	+	+	+	+	+	+	+	+	+
[25]	+	+	+	+	+	+	+	+	+
[46]	+	+	+	+	+	+	+	+	+
[28]	+	+	+	+	+	+	−	+	+
[29]	+	+	+	+	+	+	−	+	+
[40]	+	+	+	+	+	+	−	+	+
[34]	+	+	+	+	+	+	−	+	+

(+) = low risk of bias, (−) = high risk of bias.

**Table 5 ijms-25-06542-t005:** Quality of clinical trials based on Cochrane tool for assessing risk of bias in randomized clinical trials.

[Ref]	Selection Bias	Allocation Bias	Performance Bias	Detection Bias	Attrition Bias	Reporting Bias	Other Bias
[52]	−	+	+	+	?	−	?
[50]	+	+	+	+	?	?	?
[51]	+	+	+	+	?	−	?

“high −,” “low +”, or “unclear ?” risk of bias.

## Data Availability

No new data were created.

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
