# Peer review of "Emerging Therapeutic Potential of Fluoxetine on Cognitive Decline in Alzheimer’s Disease: Systematic Review"

_ijms, 2024, doi:10.3390/ijms25126542_

Round 1

Reviewer 1 Report

Comments and Suggestions for Authors

The Authors describes the activity of the antidepressant fluoxetine as neuroprotective agent for the treatment of AD.

The paper is interesting but it needs of a more in depth analysis on the activity profile of fluoxetine.

First of all the Authors cite diverse pathways (f.i. DAF-16-mediated, TGF-beta1, ILK-AKT-GSK3beta, CREB/p-CREB/BDNF) without providing a sufficient description of them. More information should be reported to make all the concepts more clear. Actually, they are too generic sentences!!!

The Authors should also make an adequate comparative analysis between the neuroprotective/antioxidant/antiinflammatory properties of fluexitine and its antidepressant activity! This aspect is only very marginally discussed, while it should be thoroughly investigated. How important is to discriminate the antidepressant effect and neuroprotective

Moreover, the aspects of toxicity need deeper and critical comments.

A lot typos occur in the paper main text. Please revise it.

The current form of the paper is too generic. An extensive elaboration of the paper main text is strongly recommended.

Comments on the Quality of English Language

English style is good.

Author Response

REVIEWER1

Comments and Suggestions for Authors

The Authors describes the activity of the antidepressant fluoxetine as neuroprotective agent for the treatment of AD.

The paper is interesting but it needs of a more in depth analysis on the activity profile of fluoxetine.

First of all the Authors cite diverse pathways (f.i. DAF-16-mediated, TGF-beta1, ILK-AKT-GSK3beta, CREB/p-CREB/BDNF) without providing a sufficient description of them. More information should be reported to make all the concepts more clear. Actually, they are too generic sentences!!!

OUR RESPONSE: Regarding DAF-16-mediated pathway, we provided more in depth analysis :’’ DAF-16 regulated the process that convert the toxic oligomers) to less-toxic aggregates [23]. In this way, fluoxetine exerts a protective effect which is further supported by the translocation of DAF-16 from cytosol to the nucleus[23]. Nevertheless, how DAF-16 mediates the protection by fluoxetine against Aβ toxicity remains obscure.

Regarding TGF-beta1, we provided more in depth analysis :’’  Fluoxetine had  neuroprotective properties against the synaptotoxic effects of Aβ oligomersAβ toxicity via a paracrine signaling mediated by increasing TGF-β1[25]. Interestingly, this study concluded that the deficit of hippocampal TGF-β1 is a long-lasting molecular marker associated with depressive like phenotype and memory deficits in a non-Tg model of AD. TGF-β1 is an anti-inflammatory cytokine that exerts neuroprotective effects in different models of amyloid-induced neurodegeneration[26].’’

ILK-AKT-GSK3β pathway: ‘’Activated Caspase-3 prevented tau phosphorylation in a transgenic AD mouse model, via the Akt- p-GSK3β levels, suggesting a potential pharmacological implication for AD[42]. Moreover, elevating ILK protein expression in the hippocampus could recover the impaired AKT-GSK3β pathway activity in APP/PS1 mice.’’

CREB/p-CREB/BDNF:’’ Fluoxetine increased increased p-CREB and BDNF levels in the hippocampus of 3×TgAD mice via the activation of the CREB/p-CREB/BDNF signalling pathway[17,20,34]. p-CREB and BDNF promoted neuroplasticity including the enhancement of hippocampal and cortex volume, increased numbers of neurons, remodelled dendritic plasticity, and increased synaptic protein levels (such as GluN2A, GluN2B, GluA1, GluA2, PSD93, PSD95, SYN1, SYT). In addition, this increased expression of BDNF may also contribute to potentiation of LTP in 3×TgAD mice. FLX administration during adolescence produced longer lasting and higher beneficial effects on the expression of BDNF and its regulatory molecules. However, the underlying mechanisms need further analysis.’’

The Authors should also make an adequate comparative analysis between the neuroprotective/antioxidant/antiinflammatory properties of fluexitine and its antidepressant activity! This aspect is only very marginally discussed, while it should be thoroughly investigated. How important is to discriminate the antidepressant effect and neuroprotective.

Moreover, the aspects of toxicity need deeper and critical comments.

OUR RESPONSE: The main aim of this review was to focus ONLY on the therapeutic potential of fluoxetine in cognitive function of AD models  exploring the anti-degenerative, anti-inflammatory and neuroprotective mechanisms and NOT the antidepressant effect of fluoxetine in AD patients. For this reason , we  applied rigorosous  exclusion criteria such as non AD-related cognitive or behavioral endpoints alone

In order to clarified the multiple neuroprotective/antioxidant/anti-inflammatory properties of fluoxetine and its antidepressant activity:” There is a crosslink of neuroprotective/antioxidant/anti-inflammatory properties and antidepressant activity of fluoxetine. Depression and AD, may share common mechanisms including neuroinflammation with an aberrant Tumor Necrosis Factor-α (TNF-α) signaling, and an impairment of Brain-Derived Neurotrophic Factor (BDNF) and Transforming-Growth-Factor-β1 (TGF-β1) signaling. The antidepressant fluoxetine  showed neuroprotection against AD rats by improving  their depressive behavior as well as their learning memory and cognition. Fluoxetine treatment of  depressive and AD animals showed that it (i) enhanced animals’ brain monoamines of 5HT, DA and NE with decrease of AChE activity; (ii)evoked neurogenesis indicated by BDNF enhancement; (iii) reduced oxidative stress manifested by upregulation of Nrf-2/HO-1 expression with increased of SOD and TAC contents along with decreased of MDA contents; (iv) decreased neuroinflammation reported by the suppression of protein expression of NF-kB/TLR4/NLRP3 pathway as it down streamed effectors, namely; TNF-α, IL-1β content besides (v) decreased of procaspase 1 protein expression. Consequently, it decreased Brain β-amyloid and Tau protein as well as it exhibited mild to moderate histopathological feature of brain histopathology as well as decreased neuronal degeneration and prevented neuronal pyknosis. Zhao et al.[55] reported the neuroprotective potential of fluoxetine against neural inflammation and apoptosis in chronic stressed-animals. Moreover, Du et al.[56]  showed neuroprotective role of fluoxetine in inhibiting NLRP3 Inflammasome activation in depressed animals. In addition, Mendez-David et al. [57] investigated the neuroprotective activity of chronic fluoxetine treatment in a mouse model of anxiety/depression via modulating Nrf2 and BDNF-signaling. These findings may open new horizons for fluoxetine as an anti-depressive drug with the potential to eliminate the risk of AD. While there is evidence that the antidepressant effect of fluoxetine requires its pro-neurogenic action, exerted by promoting proliferation, differentiation and survival of progenitor cells of the hippocampus, on the other hand fluoxetine in combination with exercice exerts also neurogenesis-independent antidepressant effects by influencing the plasticity of the new neurons generated[58].’’

Moreover, the aspects of toxicity  of fluoxetine was further described: ‘’The most frequent adverse effects, which occurred in 10–25% of patients, were nausea (25%), nervousness, insomnia, headache, tremor, anxiety, drowsiness, dry mouth, sweating, and diarrhea (10%). Most of these adverse effects occurred early in treatment and seldom led to drug withdrawal. fluoxetine worsened parkinsonian disability and akathisia because of the  enhanced serotonergic inhibition of dopamine neurons . There was an increased rate of suicide attempts. acute mania and manic-like behavior after  treatment with fluoxetine. Interestingly, cognitive function can be impaired by fluoxetine; a negative effect on learning and memory has been described.

A lot typos occur in the paper main text. Please revise it.

The current form of the paper is too generic. An extensive elaboration of the paper main text is strongly recommended.

 OUR RESPONSE: We made an extensive elaboration of the paper main text and several parts were described more in detail.

OUR RESPONSE: the manuscript was edited thoroughly by a native  English speaker

Reviewer 2 Report

Comments and Suggestions for Authors

This review evaluates the use of fluoxetine for cognitive impairment in Alzheimer's Disease (AD). The paper is well written but there a few minor concerns:

1. The introduction lacks the big picture of the work. There's enough background literature available on fluoxetine. How is this study novel?

2. The authors state "The acute administration of fluoxetine has been reported to reverse the depressive-like effect induced by Aβ1-40 administration [29]." Please discuss this in detail. What is the significance of this?

3. Since mitochondrial dysfunction is considered to be a major pathological hallmark of AD it would be informative to discuss the effect of fluoxetine on mitochondrial function (if any).

4. The authors state "By contrast, Petracca et al.[45] revealed no benefit on cognitive performance of 41 patients with PD". Do the authors mean AD or PD?

5. The authors state "The mechanisms of action of fluoxetine, including its effects on neuroplasticity, neuroinflammation, and neuroprotection, suggest that it may target key pathways involved in the pathogenesis of AD". The authors need to include Abeta accumulation, since it is one of the major key pathways that is involved. Additionally, the authors should discuss about Abeta accumulation in greater detail.

Comments on the Quality of English Language

1. There are several grammatical errors and typos. Some examples include:

A. There are various symptomatic treatment options that alleviate these symptoms only for a short time without slow down the progression of the disease.

B. By contrast, Petracca et al.[45] revealed no ebenefit on cognitive performance of 41 patients with PD.

C. Since clinical trials are lacking, the role of fluoxetine in the cognitive function in people with AD has not yet reached a consensus in cliinical practice.

D. Ninetheen studies were analyzed based on the therapeutic potential of fluoxetine in cognitive function of AD models exploring the anti-degenerative, anti-inflammatory and neuroprotective mechanisms, as following:

2. The writing style is a bit monotonous. Most of the sentences begin with either "Fluoxetine has been shown to..... " or "Fluoxetine has been found to..."

Author Response

REVIEWER 2

This review evaluates the use of fluoxetine for cognitive impairment in Alzheimer's Disease (AD). The paper is well written but there a few minor concerns:

  1. The introduction lacks the big picture of the work. There's enough background literature available on fluoxetine. How is this study novel?

OUR RESPONSE:Previous  four meta-analyses investigated the efficacy of SSRIs including fluoxetine on cognitive performance in dementia non-AD (vascular dementia,frontotemporal dementia) or.   By contrast this review is novel because it focused only on the therapeutic potential of fluoxetine in cognitive impairment in AD patients or animal models  (according strict diagnostic criteria) in order to  have a clear conclusion on cognitive effect of fluoxetine.

  1. The authors state "The acute administration of fluoxetine has been reported to reverse the depressive-like effect induced by Aβ1-40 administration [29]." Please discuss this in detail. What is the significance of this?

OUR RESPONSE: ‘’Robust Aβ1-40 immunoreactivity was verified using an anti-oligomer monoclonal antibody (NU4)6 in hippocampi from Aβ1-40-injected mice, but not in hippocampi from control vehicle-injected animals. Together, these results indicate that Aβ1-40 have an acute impact on memory, learning and mood in mice, and that fluoxetine treatment prevented both cognitive impairment and depressive-like behavior induced by Aβ1-40.’’(page 7 lines 1-7)

  1. Since mitochondrial dysfunction is considered to be a major pathological hallmark of AD it would be informative to discuss the effect of fluoxetine on mitochondrial function (if any).

OUR RESPONSE: ‘’In isolated mitochondria, fluoxetine indirectly affects electron transport and (F1Fo)ATPase activity, and thus inhibits oxidative phosphorylation, as a result of interfering with the physical state of the lipid bilayer of the inner mitochondrial membrane. Moreover, fluoxetine promoted mitophagy by increased colocalization of autophagosomes and mitochondria, eliminating damaged mitochondria in corticosterone-treated astrocytes [59]. Further in vitro study showed that p53 presence is required for fluoxetine activated autophagy flux and fluoxetine promotes astrocytic autophagy in a p53-dependent mechanism. These results may be the first step into a novel approach to treat depression depending on astrocytes, and provides a promising molecular target for the development of antidepressant drugs besides regulating neurotransmitters.’’ (page 5 lines 23-33)

  1. The authors state "By contrast, Petracca et al.[45] revealed no benefit on cognitive performance of 41 patients with PD". Do the authors mean AD or PD?

OUR RESPONSE: We corrected it : ‘’ By contrast, Petracca et al.[47] revealed no ebenefit on cognitive performance of 41 patients with AD’’

  1. The authors state "The mechanisms of action of fluoxetine, including its effects on neuroplasticity, neuroinflammation, and neuroprotection, suggest that it may target key pathways involved in the pathogenesis of AD". The authors need to include Abeta accumulation, since it is one of the major key pathways that is involved. Additionally, the authors should discuss about Abeta accumulation in greater detail.

OUR RESPONSE: ‘’Fluoxetine has been shown to modulated Aβ and tau pathology in preclinical models of AD by reducing the production of Aβ products in part via the DAF-16-mediated cell signalling pathway,[17-23] and inhibiting tau hyperphosphorylation. DAF-16 regulated the process that convert the toxic oligomers) to less-toxic aggregates [23]. In this way, fluoxetine exerts a protective effect which is further supported by the translocation of DAF-16 from cytosol to the nucleus[23]. Nevertheless, how DAF-16 mediates the protection by fluoxetine against Aβ toxicity remains obscure. Fluoxetine also reduced Aβ toxicity by activating the Wnt/β-catenin signaling, which in term down regulated the expression of BACE1 in the hippocampus of 3×Tg AD mouse[24]. This  suggests that there might be an association between neuroprotective effect of fluoxetine and the regulation of the Wnt signaling pathway in the AD brain. Fluoxetine had  neuroprotective properties against the synaptotoxic effects of Aβ oligomersAβ toxicity via a paracrine signaling mediated by increasing TGF-β1[25]. Interestingly, this study concluded that the deficit of hippocampal TGF-β1 is a long-lasting molecular marker associated with depressive like phenotype and memory deficits in a non-Tg model of AD. Fluoxetine increased the intracellular levels of the precursor of TGF-β1 (Pro-TGFβ1) without affecting TGFβ1 mRNA expression. In addition, we demonstrate that fluoxetine affected the conversion of Pro-TGF-β1 into active TGF-β1, likely through the activation of MMP-2. The involvement of MMP-2 in fluoxetine activity is in line with the suggestion that extracellular matrix modifying enzymes contribute to antidepressant-mediated structural plasticity in the hippocampus[26] TGF-β1 is an anti-inflammatory cytokine that exerts neuroprotective effects in different models of amyloid-induced neurodegeneration[27]. Four studies observed that fluoxetine-treated APP/PS1 mice and Aβ rats performed better in spatial learning, working and reference memory of APP/PS1 mice as measured by MWM, Ymaze [20,22,24,28] and Aβ rats[29]. Fluoxetine also improved more specific cortical cognitive functions memory and learning ability of cholinergic nucleus basalis Meynert (NBM) lesioned rats (experimental model of AD)[30]. The NBM has major efferent projections to medial temporal structures, the amygdala, frontoparietal cortex, and temporo-parietal association areas [31], thus stimulation of these pathways by fluoxetine might enhance cholinergic activity in these cortical regions and thereby improve mnemonic, attentional and perceptive abilities respectively. The acute administration of fluoxetine has been reported to reverse the depressive-like effect induced by Aβ1-40 administration [32]. Robust Aβ1-40 immunoreactivity was verified using an anti-oligomer monoclonal antibody (NU4)6 in hippocampi from Aβ1-40-injected mice, but not in hippocampi from control vehicle-injected animals. Together, these results indicate that Aβ1-40 have an acute impact on memory, learning and mood in mice, and that fluoxetine treatment prevented both cognitive impairment and depressive-like behavior induced by Aβ1-40. By targeting these key protein aggregates, fluoxetine may help to mitigate neuronal dysfunction and slow the progression of neurodegeneration in AD. ‘’ (pages 6, 7 lines 1-8)

Comments on the Quality of English Language

  1. There are several grammatical errors and typos. Some examples include:
  2. There are various symptomatic treatment options that alleviate these symptoms only for a short time without slow down the progression of the disease.

OUR RESPONSE: We corrected as: ‘’ There are various symptomatic treatment options that alleviate these symptoms only for a short time without slowing down the progression of the disease.’’

  1. By contrast, Petracca et al.[45] revealed no ebenefit on cognitive performance of 41 patients with PD.

OUR RESPONSE: We corrected as: ‘’ By contrast, Petracca et al.[45] revealed no benefit on cognitive performance of 41 patients with PD.’’

  1. Since clinical trials are lacking, the role of fluoxetine in the cognitive function in people with AD has not yet reached a consensus in cliinical practice.

OUR RESPONSE: We corrected as: ‘’ Since clinical trials are lacking, the role of fluoxetine in the cognitive function in people with AD has not yet reached a consensus in clinical practice.

  1. Ninetheen studies were analyzed based on the therapeutic potential of fluoxetine in cognitive function of AD models exploring the anti-degenerative, anti-inflammatory and neuroprotective mechanisms, as following:

OUR RESPONSE: We corrected as: ‘’ Nineteen studies were analyzed based on the therapeutic potential of fluoxetine in cognitive function of AD models exploring the anti-degenerative, anti-inflammatory and neuroprotective mechanisms, as following:’’

Round 2

Reviewer 1 Report

Comments and Suggestions for Authors

The Authors have notably improved the paper including more informative data about fluoxetine activity as neuroprotective agent for AD. I consider the paper of interest for Reaserchers working in the field and I suggest its publication in its current form 

Comments on the Quality of English Language

The English style is good

Reviewer 2 Report

Comments and Suggestions for Authors

The authors have addressed all my comments. The manuscript is now acceptable for publication.